# Socioeconomic Disadvantage, Residential Remoteness and Access to Specialised Interventions in Cerebral Palsy: A Cross-Sectional Study

**DOI:** 10.3390/jcm14103579

**Published:** 2025-05-20

**Authors:** Simon P. Paget, Kirsty Stewart, Lisa Copeland, Emma Waight, Nadine Smith, Felicity Baker, Jennifer Lewis

**Affiliations:** 1Children’s Hospital at Westmead Clinical School, Faculty of Medicine and Health, The University of Sydney, Sydney, NSW 2006, Australia; kirsty.stewart@health.nsw.gov.au; 2Kids Rehab, The Children’s Hospital at Westmead, Locked Bag 4001, Sydney, NSW 2145, Australia; jennifer.lewis@health.nsw.gov.au; 3Queensland Paediatric Rehabilitation Service, Queensland Children’s Hospital, Children’s Health Queensland, Brisbane, QLD 4101, Australia; lisa.copeland@health.qld.gov.au; 4Cerebral Palsy Alliance/Research Institute, Specialty of Child & Adolescent Health, The University of Sydney, Sydney, NSW 2050, Australia; emma.waight@cerebralpalsy.org.au; 5Perth Children’s Hospital, Departments of Physiotherapy and Kids Rehab, Perth, WA 6009, Australia; nadine.smith@health.wa.gov.au; 6Division of Paediatrics, Medical School, University of Western Australia, Perth, WA 6009, Australia; 7Women’s and Children’s Health Network, Adelaide, SA 5006, Australia; felicity.baker@sa.gov.au

**Keywords:** cerebral palsy, socioeconomic disparities in health, baclofen, rhizotomy, health service accessibility

## Abstract

**Aim**: Socioeconomic factors are known to influence access to health services, including for children with cerebral palsy (CP). This study aims to determine whether socioeconomic disadvantage and/or geographical remoteness influence access to specialised CP interventions: selective dorsal rhizotomy (SDR) and intrathecal baclofen (ITB). **Methods**: This was a cross-sectional study of children with CP from (i) the Australian SDR Research Registry and (ii) an Australian ITB audit study. Socioeconomic disadvantage was grouped (quintiles) using the Index of Relative Socioeconomic Disadvantage (IRSD). Geographical remoteness was determined using the Australian Statistical Geographical Standard. IRSD quintiles and remoteness were compared with the Australian CP Register (ACPR) (birth years 1995–2016). **Results**: A total of 64 children (31.3% female) had received SDR surgery and 52 children (48.1% female) had received ITB therapy. Of these, 7 (11.1%) (SDR) and 7 (13.5%) (ITB) lived in the most disadvantaged neighbourhoods (IRSD quintile 1); 41 children (65.1%) (SDR) and 42 (82.4%) (ITB) lived in major cities. In comparison, 1630 (18.8%) of children on the ACPR resided in IRSD quintile 1; 6122 (70.4%) resided in major cities. There were no statistical differences in IRSD distribution between ACPR, SDR, and ITB groups. More children in major cities received ITB therapy (*p* = 0.03) and more children in outer regional/remote areas had received SDR (*p* = 0.03). **Conclusions**: Access to SDR and ITB in Australia varies by geographical remoteness. Equity of access is important to monitor, and interventions should be considered to reduce inequity.

## 1. Introduction

Cerebral palsy (CP) is the most common physical disability in childhood, with a current birth incidence in Australia of 1.4 per 1000 live births [1]. CP is clinically diverse, representing a range of genetic and environmental aetiologies that disrupt early brain development and cause disorders of movement and posture of different types and severities [2].

CP is typically described in terms of biological mechanisms, but there is increasing understanding that socioeconomic factors also play an important role in the condition [3]. Socioeconomic disadvantage has been shown to be associated with increased CP severity (gross motor function impairment, severity of intellectual disability, and severe comorbidities (epilepsy, functional blindness, bilateral deafness, no verbal communication)) for children living in high-income countries [4,5]. Socioeconomic factors can also influence access to health services for children with CP. Socioeconomically disadvantaged children with developmental vulnerabilities are less likely to access primary, specialist, and hospital health care services than children from less disadvantaged backgrounds [6]. Children with CP who are at greater socioeconomic disadvantage are also more likely to miss scheduled hospital outpatient appointments [7]. A recent study has identified differences in access to a specialised intervention (intrathecal baclofen (ITB)) in Europe, influenced by the economic health (gross domestic product) of individual countries [8]. Less is known about whether socioeconomic factors impact access to specialised CP interventions such as selective dorsal rhizotomy (SDR) and ITB in Australia.

Due to their specialised nature, ITB and SDR surgeries are restricted to several of the tertiary metropolitan children’s hospitals in Australia (ITB in six hospitals in five states; SDR in four hospitals in four states). Family factors are generally regarded as being important in selection for both procedures due to the substantial commitment for prolonged and intensive rehabilitation after SDR surgery, and the need for regular baclofen refills and/or troubleshooting in ITB [9,10]. Almost 30% of children with CP in Australia live in regional or remote areas [11], and only a small minority (approximately 9%) of families of children with CP move to less remote areas to access services during the first 5 years of life [12]. Children who live in regional and remote areas of Australia are recognised to have poorer access to health services [13].

The aim of this study was to examine whether socioeconomic disadvantage and/or geographical remoteness influence families’ access to SDR surgery and ITB therapy in Australia.

## 2. Materials and Methods

We performed a cross-sectional study of children with CP who had received either SDR surgery or ITB therapy. The study setting was Australia. The Australian health system is complex and includes services funded by the Australian (Federal) Government, state and territory governments, and through the private sector [14]. SDR and ITB services for children with CP are almost exclusively provided at major specialist paediatric public hospitals, and are generally provided free at the point of contact, managed largely by state and territory governments, and supported through funding provided by the Australian Government [15].

Children who had had SDR surgery were identified from the Australian SDR Research Registry (trial registration number ACTRN12618000985280) [16]. The Australian SDR Research Registry collects data on children who have had SDR surgery (in Australia and overseas), recruiting families from the rehabilitation services at five major children’s hospitals in Australia. We identified children who had received ITB therapy from the Intrathecal Baclofen therapy for children: Australian paediatric multicentre study (trial registration number ACTRN 12610000323022). This is a longitudinal, prospective, multi-centre national audit involving six children’s hospitals in Australia [17]. We excluded children from analyses where (i) surgery had occurred in countries other than Australia or (ii) children had diagnoses other than CP. Ethics approval for the Australian SDR Registry was obtained from the Sydney Children’s Hospital Network, 2022/ETH02325 (NSW, SA, VIC, QLD) and PRN: RGS0000000323 (WA). National ethics approval for Intrathecal Baclofen therapy for children: an Australian multicentre study was obtained through Children’s Health Queensland HREC/16/QRCH/391.

Data retrieved from these databases included demographics (sex, age, age at surgery), functional classifications (Gross Motor Function Classification System (GMFCS) [18], Manual Abilities Classification System (MACS) [19], Communication Function Classification System (CFCS) [20]), and postcode of residence. Postcode of residence was used to determine socioeconomic disadvantage and geographical remoteness. Socioeconomic disadvantage was derived from the Index of Relative Socioeconomic Disadvantage (IRSD) [21]. IRSD is a general socioeconomic index that summarises the economic and social conditions of households at an area level, and includes indicators such as household income, education level, employment, vehicle ownership, and (English) language proficiency. IRSD rank scores were grouped into (national) quintiles (quintile 1 most disadvantaged and quintile 5 least disadvantaged). Geographical remoteness was defined using the Australian Statistical Geography Standard [22], which categorises populated localities as major cities, inner/outer regional, and remote areas based on ease of access to services via the road network. We combined outer regional and remote areas because of the small number of children identified as living in remote areas.

We compared the Index of Socioeconomic Disadvantage quintiles and geographical remoteness of children identified in the (ITB/SDR) specialised intervention groups with population data from (*n* = 10,857) children with CP in the Australian Cerebral Palsy Register (ACPR) born in Australia in 1995–2016. The ACPR is a confidential research database that holds demographic, perinatal, and clinical data of individuals that have been identified as having CP in Australia [11]. The ACPR consolidates de-identified information from the CP Registers every two years from each of the eight Australian states’ and territories’ registers. Postcode of residence for children on the ACPR was obtained at 5 years of age, or at birth where data for the age of 5 years were missing.

Data are described using frequencies and proportions for categorical data. Continuous data are described using medians and interquartile ranges (given the data’s skewed distribution). Comparisons of the distributions of IRSD quintiles and geographical remoteness categories between specialised intervention groups and ACPR were performed using chi-squared or Fisher’s exact test (where expected cell counts < 5). Consistent with a previous study [23], the distribution of socioeconomic disadvantage in the ACPR differed by GMFCS group (Figure 1) (chi-squared *p* = 0.02). We therefore performed sensitivity analyses comparing socioeconomic disadvantage between intervention groups and subgroups of the ACPR based on GMFCS groups similar to the intervention groups. We compared the SDR cohort to children GMFCS I-III in the ACPR, and compared children in the ITB cohort to children GMFCS IV-V in the ACPR. Analyses were conducted using SAS 9.4 (SAS Institute, Cary, NC, USA).

## 3. Results

We identified 64 children with CP (*n* = 20, 31.3% female) who had received SDR surgery (birth years 1997–2019; surgery dates 2003–2024; *n* = 20, 31.3% travelled interstate for SDR surgery) and 52 children with CP (*n* = 25, 48.1% female) who had received ITB therapy (birth years 1995–2017; surgery dates 2010–2023). Median age at SDR surgery was 6.1 years (interquartile range (IQR) 5.2–6.9 years), and for the ITB implant it was 11.1 years (IQR 8.3–13.1 years). Clinical details of both patient groups are shown in Table 1. Intervention groups differed by distribution of functional classifications (mode GMFCS: III (*n* = 35) for SDR, V (*n* = 33) for ITB; mode MACS: I (*n* = 35) for SDR, V (*n* = 30) for ITB; mode CFCS I (*n* = 52) for SDR, V (*n* = 24) for ITB).

Children in both intervention groups were identified in each IRSD quintile. Among the children who had received SDR surgery, *n* = 17 (27.0%) children lived in the least disadvantaged areas (IRSD quintile 5), and *n* = 7 (11.1%) lived in the most disadvantaged areas (IRSD quintile 1). Among the children who had received ITB therapy, *n* = 11 (21.1%) lived in the least disadvantaged areas (IRSD quintile 5), and *n* = 7 (13.5%) lived in the most disadvantaged areas (IRSD quintile 1). Comparison of intervention groups with the ACPR is shown in Figure 2. Compared with the ACPR (18.8%), quintile 1 was under-represented in both intervention groups (SDR 11.1%, ITB 13.5%). There was no statistical evidence of differences in distributions of IRSD quintiles between the SDR registry and ACPR (*p* = 0.27) or the ITB cohort and ACPR (*p* = 0.23). Sensitivity analyses comparing distributions of IRSD quintiles between the SDR group and ACPR (GMFCS I-III), and the ITB group and ACPR (GMFCS IV-V), showed no statistical evidence of differences.

Most children who had received SDR surgery or the ITB implant lived in major cities (SDR: 65.1%, *n* = 41; ITB 82.4%, *n* = 42) or inner regional areas (SDR: 14.3%, *n* = 9; ITB 5.9%, *n* = 3). Comparison of intervention groups with geographical remoteness of the ACPR population is shown in Figure 3. Compared with the ACPR, there was a higher proportion of children who had received SDR surgery who lived in outer regional and remote areas (SDR 20.6%, *n* = 13 v ACPR 10.5%, *n* = 914; *p* = 0.03). Compared with the ACPR, there was a higher proportion of children who had received the ITB implant who lived in major cities (ITB 82.4%, *n* = 42 v ACPR 70.4%, *n* = 6122; *p* = 0.03).

## 4. Discussion

We found that children with CP living in the most socioeconomically disadvantaged postcodes were under-represented among children who had received SDR or ITB implant surgeries. However, there was no statistical evidence of a difference in socioeconomic disadvantage distribution between specialised intervention groups and the whole CP population, identified in the ACPR. Children living in major cities were over-represented amongst those with ITB implants, and, conversely, children living in outer regional and remote areas were more likely to have received SDR surgery.

The lack of an overall socioeconomic gradient in access to specialised interventions for CP in Australia is pleasing, but our finding of reduced access to both interventions for children living in the most socioeconomically disadvantaged postcodes requires further exploration. The existence of a social gradient of health for children living with socioeconomic disadvantage experiencing multiple worse health outcomes is well established [24,25,26]. This includes children with CP, where children from more socioeconomically disadvantaged neighbourhoods in Australia are more likely to have severe CP, intellectual disability, and epilepsy [4]. Health outcomes for those with socioeconomic disadvantage can be compounded by increased barriers to accessing health care—Tudor Hart’s Inverse Care law [27]. Whilst there is no evidence of significant differences in total access to public hospital outpatient clinics for children and young people with CP related to socioeconomic disadvantage [28], children with socioeconomic disadvantage are less likely to experience high continuity of care [28] and are more likely to miss scheduled outpatient clinics [7]—differences which may impact referrals for intensive interventions such as SDR and ITB.

The increased rates of ITB implants in major cities may represent the ‘tyranny of distance’, recognising the challenges of travelling to tertiary paediatric centres and/or costs for families in regional and remote Australia that present substantial barriers to accessing health care [13]. There are known logistic challenges associated with the requirement for regular refills with ITB therapy (which can be as frequently as every 4–8 weeks), and unexpected or unplanned hospital admissions and appointments for troubleshooting if pump malfunction is suspected. The finding of increased SDR in outer regional and remote areas is more surprising, but may be related to decision-making also influenced by geographical remoteness. Australia’s geography, and the concentration of specialised health care in major cities, means that families living in outer regional and remote areas are required to travel for many hours to receive health care interventions such as botulinum toxin A. In this circumstance, clinicians may be inclined to suggest (more invasive) single-event procedures (e.g., SDR) rather than interventions that require substantial and repeated travel (e.g., botulinum toxin A) where travel is onerous.

While SDR and ITB in Australia are typically offered through the public health system with no out-of-pocket costs for patients and families, the surgery and requirements for intensive and prolonged rehabilitation (SDR) or regular refills (ITB) may have substantial impact on employment and caring roles for other children [29]. Families who experience more socioeconomic disadvantage may find this impact to be a barrier to proceeding with these interventions, even when clinically indicated. Clinicians may also be less likely to discuss or suggest these interventions to families who experience socioeconomic disadvantage because of implicit or explicit beliefs (common in health care professionals) that the family is unable to contend with the demands of the procedure [30]. Access may also be influenced by differences in awareness about SDR and ITB, impacted by differences in health literacy and the availability of knowledgeable community teams, which may be a particular challenge in more remote settings [31]. In this context, more accessible information about these specialised procedures is needed to improve awareness [32].

This study has strengths in its use of national patient registries and comparison with data from the ACPR. We are confident that the SDR Registry contains almost all children who have had the surgery in Australia. Limitations include differences in completeness in data from different Australian states in the ITB cohort study. The small number of children in both datasets may also have influenced the results of the statistical tests conducted. We also did not include children who had travelled overseas for SDR surgery. Whilst the focus of this study was the Australian health system, these children are an important group to consider.

Access to high-quality health care is an important determinant of health outcomes, including for children with CP. This study suggests that access to specialised interventions may be influenced by geographical remoteness and socioeconomic disadvantage. Further work will be required to explore and eliminate any unintended clinical variation that results in these health inequities.

## Figures and Tables

**Figure 1 jcm-14-03579-f001:**
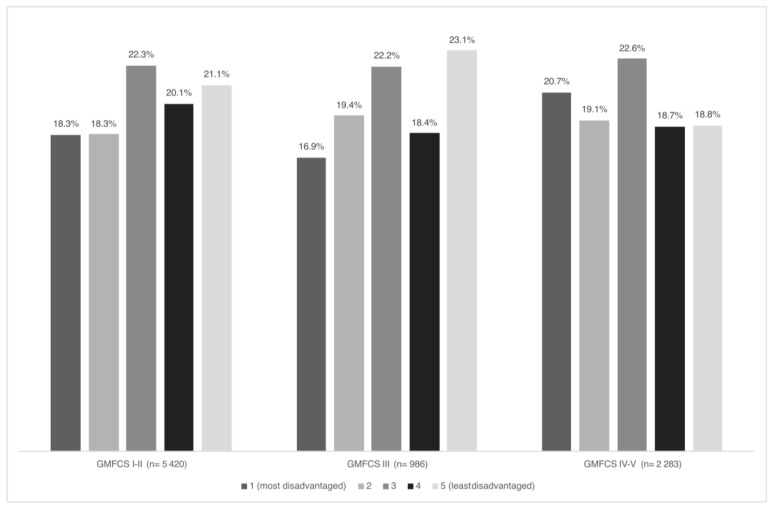
Distribution of socioeconomic disadvantage (Index of Relative Socioeconomic Disadvantage) in the Australian Cerebral Palsy Register (birth years 1995–2016).

**Figure 2 jcm-14-03579-f002:**
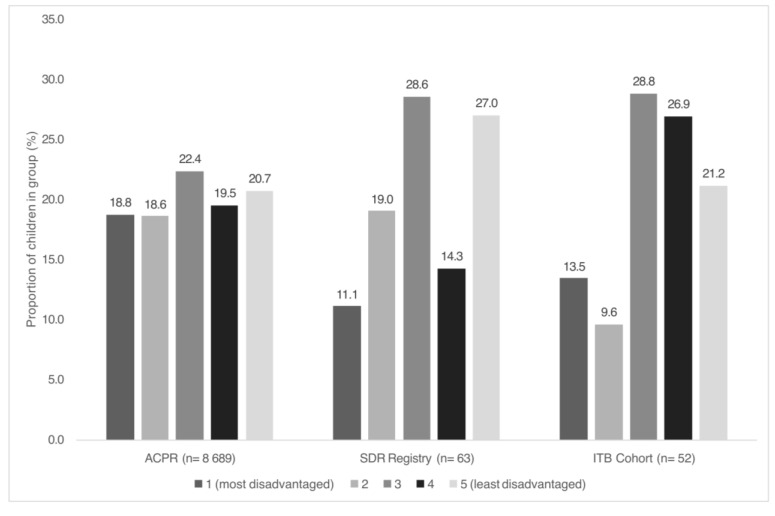
Comparison of Index of Relative Socioeconomic Disadvantage (quintiles) for children with cerebral palsy who have undergone selective dorsal rhizotomy surgery or intrathecal baclofen pump implant in the Australian Cerebral Palsy Register. ACPR Australian Cerebral Palsy Register; ITB intrathecal baclofen; SDR selective dorsal rhizotomy.

**Figure 3 jcm-14-03579-f003:**
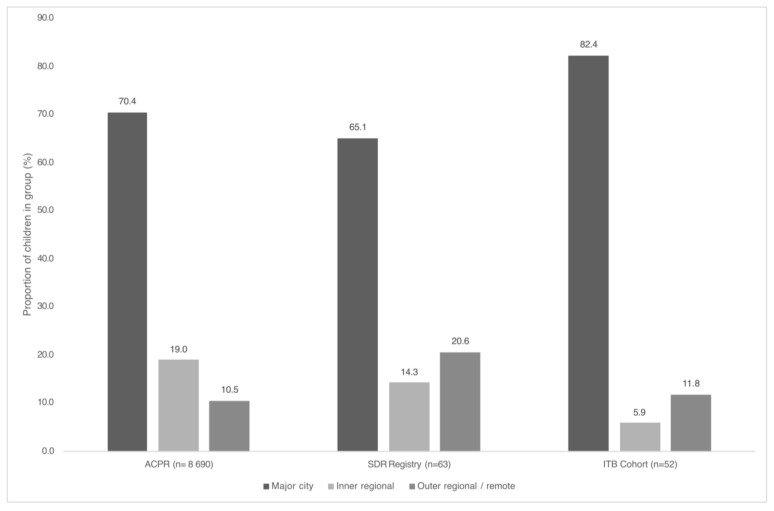
Comparison of geographical remoteness for children with cerebral palsy who have undergone selective dorsal rhizotomy surgery or intrathecal baclofen pump implant in the Australian Cerebral Palsy Register. ACPR Australian Cerebral Palsy Register; ITB intrathecal baclofen; SDR selective dorsal rhizotomy.

**Table 1 jcm-14-03579-t001:** Functional classifications of children with cerebral palsy in the Australian Selective Dorsal Rhizotomy Registry or Intrathecal Baclofen Cohort Study.

	SDR	ITB
Total (*n*)	64	52
Sex *n* (%)		
Male	44 (68.8)	27 (51.9)
Female	20 (31.3)	25 (48.1)
GMFCS *n* (%)		
I		
II	21 (33.3)	
III	35 (55.6)	1 (1.9)
IV	7 (11.1)	18 (34.6)
V		33 (63.5)
MACS *n* (%)		
I	35 (56.5)	
II	23 (37.1)	2 (3.8)
III	4 (6.5)	8 (15.4)
IV		12 (23.1)
V		30 (57.7)
CFCS *n* (%)		
I	52 (86.7)	3 (5.9)
II	4 (6.7)	5 (9.8)
III	4 (6.7)	9 (17.6)
IV		10 (19.6)
V		24 (47.1)

CFCS Communication Function Classification System; GMFCS Gross Motor Function Classification System; ITB intrathecal baclofen; MACS Manual Abilities Classification System; SDR selective dorsal rhizotomy.

## Data Availability

The original contributions presented in this study are included in the article. Further inquiries can be directed to the corresponding author.

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
