# Peer review of "Socioeconomic Disadvantage, Residential Remoteness and Access to Specialised Interventions in Cerebral Palsy: A Cross-Sectional Study"

_jcm, 2025, doi:10.3390/jcm14103579_

Round 1

Reviewer 1 Report

Comments and Suggestions for Authors

Well done to the authorship team on a well written, well designed and interesting peice of work. My comments are only very minor.

  • Abstract: within the methods, are you missing an 'and' between "the Registry, and ii) an Australian..."
  • Introduction: in the final paragraph, t may be worth stating how many hospitals provide ITB and SDR in Australia (i.e. being more specific than 'several').
  • Methods: It was noted that children who had SDR overseas were excluded, which I agree is suitable to do, however it may be pertinent to provide a abrief description or commentary on that group of children (if that data are available? though I do recognise this could be extended beyond the scope of this study). Perhaps at least this might be noted within the Discussion as a talking point? 
  • Methods/Results: just missing the Figure and Table numbers within text
  • Figures: it may be worth including the n or the sample within the bars or figure captions, and the % at the end of the bars for figure 2 and 3. 
  • Is it perhaps also worth noting if (/how many?) children had travelled interstate to have SDR?
  • Discussion: Well considered discussion, great job. Perhaps noting the limitation around families choosing to travel Overseas to complete SDR? 

Author Response

Comment 1: Abstract: within the methods, are you missing an 'and' between "the Registry, and ii) an Australian..."
Response 1: Thank you for identifying this mistake - it has been corrected (Abstract, Methods).

Comment 2: Introduction: in the final paragraph, it may be worth stating how many hospitals provide ITB and SDR in Australia (i.e. being more specific than 'several').
Response 2: Thank you. We have added this additional information "(ITB in 6 hospitals in 5 states, SDR in 4 hospitals in 4 states)".

Comment 3: Methods: It was noted that children who had SDR overseas were excluded, which I agree is suitable to do, however it may be pertinent to provide a abrief description or commentary on that group of children (if that data are available? though I do recognise this could be extended beyond the scope of this study). Perhaps at least this might be noted within the Discussion as a talking point?
Response 3: Thank you. We have added the following commentary to the limitations section in the Discussion: "We also did not include children who had travelled overseas for SDR surgery. Whilst the focus of this study was the Australian health system, these children are an important group to consider."

Comment 4: Methods/Results: just missing the Figure and Table numbers within text
Response 4: Thank you and apologies. These must have disappeared during formatting? We have added them.

Comment 5: Figures: it may be worth including the n or the sample within the bars or figure captions, and the % at the end of the bars for figure 2 and 3.
Response 5: We have updated figures 2 and 3 to include this information.

Comment 6: Is it perhaps also worth noting if (/how many?) children had travelled interstate to have SDR?
Response 6: We have included this information in Results "We identified 64 children with CP (n=20, 31.3% female) who had SDR surgery (birth years 1997-2019; surgery dates 2003-2024; n=20, 31.3% travelled interstate for SDR surgery)"

Comment 7: Discussion: Well considered discussion, great job. Perhaps noting the limitation around families choosing to travel Overseas to complete SDR?
Response 7: Thank you and changes noted as above in Comment 3.

Reviewer 2 Report

Comments and Suggestions for Authors

This is a nice study looking at the impact of socio-economic status and geographical location on referral for SDR and ITB surgeries.

Below are some minor comments about the manuscript. 

Materials and Methods
Page 3 when referencing your figure, you forgot to number figure 1.

Your data analysis paragraph is a bit difficult to follow. For example, the sentence: "Comparison of the distributions of IRSD quintiles and geographical remote-ness categories between specialised intervention groups and ACPR". What do you mean by ACPR? From what I understand ACPR is a database so what exactly in the database are you comparing?
The next sentence is more of a result in which you sorted the % of children by IRSD and GMFC level. Although it is unclear why you grouped GMFCS I-II and GMFCS IV-V together and how many children are in each GMFCS group.
Then you mention sensitivity analysis but I did not understand exactly what you did in that sensitivity analysis. Would you be able to develop a bit more on this sensitivity methodology?

I would actually put the ethics on top of the method section rather than bottom.

You mention that you extracted data for 10,857 children. The definition of children is a bit vague so it would be nice if you could expand on the age range that you targeted in the extraction and maybe inclusion and exclusion criteria if you had any.

Results
You should think about numbering your tables and figures when referenced in the text, it would make it easier on the reader.

Page 6, first sentence is not a sentence but should I believe belong to figure 2 caption? Same with the first sentence in page 7 and figure 3 caption.

Author Response

Comment 1: Page 3 when referencing your figure, you forgot to number figure 1.
Response 1: Thank you and apologies. These must have disappeared during formatting? We have added 

Comment 2: Your data analysis paragraph is a bit difficult to follow. For example, the sentence: "Comparison of the distributions of IRSD quintiles and geographical remote-ness categories between specialised intervention groups and ACPR". What do you mean by ACPR? From what I understand ACPR is a database so what exactly in the database are you comparing?
Response 2: Thank you. We've adjusted this section to help with clarity. It now opens: "We compared Index of Socioeconomic Disadvantage quintiles and geographical remoteness of children identified in the specialised intervention groups with population data from (n=10,857) children with CP in the Australian Cerebral Palsy Register (ACPR) born in Australia 1995-2016." The ACPR is introduced in this paragraph: "The ACPR is a confidential research database that holds demographic, perinatal and clinical data of individuals that have been identified as having CP in Australia.(11) The ACPR consolidates de-identified information from the CP Registers every two years from each of the eight Australian states and territories registers. Postcode of residence for children on the ACPR was obtained at age 5 years, or at birth where age 5 years data were missing." 

Comment 3: The next sentence is more of a result in which you sorted the % of children by IRSD and GMFC level. Although it is unclear why you grouped GMFCS I-II and GMFCS IV-V together and how many children are in each GMFCS group.
Response 3: Thank you. Our preference is to leave this description here (in Materials and Methods) as it describes the population database that we compare our speciality intervention (SDR, ITB) cohorts with (rather than being a result). Data from the ACPR were available to us grouped by the GMFCS levels that we report. We feel it is helpful to report in this way as it supports the sensitivity analysis that enables comparison of the SDR cohort with children in the ACPR (GMFCS I-III) and ITB cohort with ACPR (GMFCS IV-V). We have added the number of children in each GMFCS group to figure 1.

Comment 4: Then you mention sensitivity analysis but I did not understand exactly what you did in that sensitivity analysis. Would you be able to develop a bit more on this sensitivity methodology?
Response 4: Thank you. We have added this additional information which we hope is helpful: "We compared the SDR cohort to children GMFCS I-III in the ACPR, and compared children in the ITB cohort to children GMFCS IV-V in the ACPR."

Comment 5: I would actually put the ethics on top of the method section rather than bottom.
Response 5: Thank you. We have moved the ethics approval information to the end of paragraph 2 in materials and methods.

Comment 6: You mention that you extracted data for 10,857 children. The definition of children is a bit vague so it would be nice if you could expand on the age range that you targeted in the extraction and maybe inclusion and exclusion criteria if you had any.
Response 6: Thank you. The 10,857 children were a population comparison group of children from the Australian Cerebral Palsy Register (ACPR). The text describes these as "children with CP in the Australian Cerebral Palsy Register (ACPR) born in Australia 1995-2016." We hope this is sufficient description?

Comment 7: You should think about numbering your tables and figures when referenced in the text, it would make it easier on the reader.
Response 7: Thank you and apologies. These must have disappeared during formatting? We have added them.

Comment 8: Page 6, first sentence is not a sentence but should I believe belong to figure 2 caption? Same with the first sentence in page 7 and figure 3 caption.
Response 8: Thank you. I have highlighted these are legends for the figures and not part of the main text.

Reviewer 3 Report

Comments and Suggestions for Authors

Thank you for allowing me to review this research. The manuscript is easy to read, and the references used are appropriate for the topic.

I would like to make some suggestions for improving the manuscript.
- The ORCIDs in the researchers' names are not included there.
- Methods. Results are being provided; they should be included in the results section.
- Results. Review the figure names because they do not specify whether they are 1, 2, or 3.
- A conclusions section is missing.

Author Response

Comment 1: The ORCIDs in the researchers' names are not included there.
Response 1: These have been added.

Comment 2: Methods. Results are being provided; they should be included in the results section.
Response 2: Thank you. We consider these not to be results of the study, as they provide contextual information about the population dataset that provides the comparison with our cohorts. We humbly suggest that this information remains where it is. 

Comment 3; Results. Review the figure names because they do not specify whether they are 1, 2, or 3.
Response 3: Thank you and apologies. These must have disappeared during formatting? We have added them.

Comment 4: A conclusions section is missing.
Response 4: Thank you. Our reading of Instructions for Authors is that the Conclusions section is optional? We suggest that the discussion remain as written; but happy to discuss further.